# Easy, Flexible and Standardizable Anti-Nascent Biofilm Activity Assay to Assess Implant Materials

**DOI:** 10.3390/microorganisms11041023

**Published:** 2023-04-14

**Authors:** Jérome Vanheuverzwijn, Eloise-Eliane Maillard, Amal Mahat, Lee Fowler, Daniel Monteyne, Leïla Bonnaud, Nicolas Landercy, Axel Hemberg, Ana Janković, Franck Meyer, Vesna Mišković-Stanković, Milena Stevanović, Codruta Mirica, David Pérez-Morga, Reto Luginbuehl, Christèle Combes, Gabriel Furtos, Véronique Fontaine

**Affiliations:** 1Microbiology, Bioorganic and Macromolecular Chemistry Unit, Faculty of Pharmacy, Université libre de Bruxelles (ULB), Boulevard du Triomphe, 1050 Brussels, Belgium; 2Applied Materials Science, The Ångström Laboratory, Department of Engineering Sciences, Uppsala University, P.O. Box 534, 75121 Uppsala, Sweden; 3Laboratoire de Parasitologie Moléculaire, Faculté des Sciences & CMMI, Université Libre de Bruxelles (ULB), CP 300. Rue Prof. Jeener & Brachet, 12, 6041 Gosselies, Belgium; 4Center of Innovation and Research in Materials and Polymers, Materia Nova Research Center & University of Mons, 7000 Mons, Belgium; 5Faculty of Technology and Metallurgy, University of Belgrade, Karnegijeva 4, 11000 Belgrade, Serbia; 6Faculty of Ecology and Environmental Protection, University Union-Nikola Tesla, Cara Dusana 62-64, 11158 Belgrade, Serbia; 7Department of Oral Health, Iuliu Hatieganu University of Medicine and Pharmacy, Victor Babes Street 15, 400012 Cluj-Napoca, Romania; 8Department of Biomedical Material Research, University of Bern, 3008 Bern, Switzerland; 9Blaser Swisslube, 3415 Hasle-Rüegsau, Switzerland; 10Centre Inter-Universitaire de Recherche et d’Ingénierie des Matériaux, CIRIMAT, Toulouse INP, Université Toulouse 3 Paul Sabatier, CNRS, Université de Toulouse, 4 allée Emile Monso, BP44362, CEDEX 4, 31030 Toulouse, France; 11Department of Dental Materials, Institute of Research in Chemistry, Babes-Bolyai University-Raluca Ripan, Fantanele Street 30, 400294 Cluj-Napoca, Romania

**Keywords:** surface, anti-biofilm, infection, medical device, copper, metal ion

## Abstract

Medical implants have improved the quality of life of many patients. However, surgical intervention may eventually lead to implant microbial contamination. The aims of this research were to develop an easy, robust, quantitative assay to assess surface antimicrobial activities, especially the anti-nascent biofilm activity, and to identify control surfaces, allowing for international comparisons. Using new antimicrobial assays to assess the inhibition of nascent biofilm during persistent contact or after transient contact with bacteria, we show that the 5 cent Euro coin or other metal-based antibacterial coins can be used as positive controls, as more than 4 log reduction on bacterial survival was observed when using either *S. aureus* or *P. aeruginosa* as targets. The methods and controls described here could be useful to develop an easy, flexible and standardizable assay to assess relevant antimicrobial activities of new implant materials developed by industries and academics.

## 1. Introduction

Over the last 40 years, innovations in electronics, functional materials and implant designs have led to the development of various medical devices, such as heart valves, endovascular stents and joint prostheses, targeting various tissues. The utilization of medical devices and implants has become prevalent in health treatments within countries with higher levels of income. Moreover, recent innovative medical devices, such as artificial pancreas device systems, artificial hearts or deep brain stimulators, could become more commonplace [1,2,3]. While medical implants have improved the survival and quality of life of many patients, a highly variable, but significant percentage of these suffer from healthcare-associated infections (HAIs). For instance, the incidence rate is less than 5% for dental implants, central venous catheters or prosthetic joints, but it is at more than 25% for heart valves [3]. The consequence of HAIs, in any case, is medical therapy, but it may lead to repeated surgical treatments and device replacements.

Biofilm formation on sterile medical implants and devices subjected to microbial contamination is the major cause of these infections [2,3]. This biofilm formation follows four stages: (1) adhesion to the material, (2) micro colony development, (3) biofilm maturation and (4) microorganism dispersion. It arises either by unique microbial colonization or multiple microorganism colonizations [4,5]. The interaction between microorganisms and surfaces depends on the material’s physiochemical properties, particularly surface charges [4,5]. After adhesion and micro-colonization, microorganisms embed themselves in secreted extracellular polymeric substances (EPSs). Microorganisms that are engulfed in EPSs might encounter hypoxic conditions or nutrient deprivations, changing their metabolisms and leading them to enter a lag phase of dormancy and non-replicative state. These bacteria are consequently the most recalcitrant to antimicrobial treatment by tolerance, but also the most prone to acquired resistance [5]. Secreted metabolites produced by some microorganisms can be nutrients or attachment points for other microorganisms. When microbial density reaches a critical survival point (quorum), microorganism gene expression is modified at the transcriptional level by a quorum sensing system (involving auto-inducers involved in inter-microorganism communication) [5]. This leads to biofilm perturbation and the dispersion of microorganisms [5]. Although new antimicrobial surfaces are being developed, current testing methods for evaluating engineered antimicrobial surfaces do not adequately quantify their anti-biofilm activities and bacterial contamination in a standardized manner that reflects bacterial contamination, especially regarding their anti-biofilm activities [2,3,4]. For medical devices and implants, the goal of their antimicrobial activities is to fight against the potential attachment of a microorganism (anti-adherent activity) or to kill microorganism (biocidal activity) that could bind the surface during surgery. Indeed, targeting mature biofilms by killing engulfed microorganisms and dispersing the secreted microbial compounds, including extracellular polymeric substances, is not only a difficult task, but above all, an unnecessary challenge [2,3]. The ISO 22196 international standard procedure, also called the film-covering method, derived from the Japanese standard JIS Z2801, allows for assessing the antibacterial properties of materials [2,4,5]. However, it grants limited flexibility in terms of shear stress conditions, and no information on material properties to inhibit biofilm development [2,5]. When assessing biofilm formation, it is tempting to quantify biofilm by an easy and rapid staining method, such as crystal violet [4,5,6]; however, this usually requires additional testing to obtain reliable quantitative assessment not only of the biofilm biomass, but also of the viable microorganisms [4,5,6]. Nevertheless, each procedure or method offers advantages and disadvantages [5,7,8].

The aim of this research was to develop an easy, flexible and standardized antibiofilm activity assay on nascent biofilms (biofilm development inhibition assay) to assess anti-nascent biofilm activity for the evaluation of potential surface materials for implants. Since antimicrobial activities of some metal ions are well-established [9], our study investigated the potential of Euro coins as positive antimicrobial surface control.

## 2. Materials and Methods

### 2.1. Chemicals

The chemicals used in our study included poly(ε-caprolactone) (PCL) (80,000 g.mol^−1^ M.W.), glutaraldehyde, polysorbate 80, chloroform, methanol and hexamethyldisilazane, and were purchased from Sigma-Aldrich GmbH, Steinheim, Germany. Tetracycline (TET) was supplied by Antibiotice SA Iasi, Romania. Poly(lactic acid) (PLA) Ingeo™ 4032D produced by NatureWorks LLC, in pellet shape, was used [10,11]. PLA had a film extrusion grade with the D-isomer of 1.4%, a relative viscosity of 3.94 and contained 0.14% residual monomers.

### 2.2. Surfaces

The used surfaces were round, flat and at a size of a 5 Euro cent coin (2 cm in diameter, 3 mm in thickness).

#### 2.2.1. Control Surfaces

We used borosilicate glass tokens (Pierre E. sprl, Vilvoorde, Belgium) as the negative control surface (without antimicrobial activity), and 5 Euro cent coins as the positive control surface (with antimicrobial activities).

#### 2.2.2. Material Surface Manufacturing


**Metal surface manufacturing**


Cu and Zn test coupons were produced by punching 2 cm diameter samples out of a 3 mm sheet metal (Metallica Sàrl, Crissier, Switzerland).


**Polycaprolactone membrane with tetracycline**


Two grams of PCL with 0.5 wt.% or 1 wt.% tetracycline (TET) were dissolved in 10 mL chloroform/methanol mixture (3:1 *v/v*) and cast onto glass petri dishes until solvent evaporation. The resulting films were punched to obtain 2 cm diameter circular holes for the samples.


**Silver-loaded cements**


Reference and Ag-loaded cements were prepared following a published protocol [12]. Briefly, a reference cement paste was prepared by mixing the solid phase (a powder mixture of synthesized dicalcium phosphate dihydrate, CaHPO_4_. 2H_2_O, vaterite, CaCO_3_) with the liquid phase (deionized water), using a liquid/solid weight ratio of 0.7. In the case of silver-loaded cements, the antibacterial agent, namely the silver ion, was incorporated in the cement powder mixture (solid phase) as a synthetic silver phosphate Ag_3_PO_4_ powder [12], at three concentrations: 0.0375 wt.%, 0.375 wt.% and 1 wt.% of Ag in the solid phase, before mixing with deionized water. After manual mixing, the pastes were filled into cylindrical silicone moulds (diameter: 11 mm, height: 4 mm), placed in sealed containers and left at 37 °C in a water-saturated atmosphere for 48 h, for setting and hardening. The cement pellets (n = 5 for reference and each Ag-loaded cement composition) were unmoulded and then dried at 37 °C for two days. The hardened and dried cements were mainly composed of biomimetic apatite and some remaining unreacted vaterite. The amount of silver phosphate in the cement was too low to be detected by X-ray diffraction analysis even for the higher concentration of Ag_3_PO_4_ tested (1 wt.% of Ag) [12].


**Monolayered (PLA/Ag or PLA/ZnOAg) and three-layered ZnO PLA surfaces (PLA/3xZnO and PLA/3xZnO/ZnOAg)**


Ag thin layer (10 nm thickness, PLA/Ag) and Ag-doped ZnO thin layers were deposited on organic PLA-based films by physical vapor deposition, following a magnetron sputtering coating process, as previously described [13]. Here, the thickness of the deposited thin layers is 5, 20 and 80 nm (PLA/5ZnOAg, PLA/20ZnOAg, PLA/80ZnOAg and PLA/3xZnO/5ZnOAg, PLA/3xZnO/20ZnOAg, PLA/3xZnO/80ZnOAg).

For the production of all films, Ultranox 626A (Bis (2,4-di-t-butylphenyl) pentraerythritol diphosphite), a powder supplied by Brenntag NV, was used as a thermal stabilizer, at a percentage of 0.25% in all PLA compositions designed for the production by the extrusion of films.

In brief, for the production of monolayer films, PLA pellets were dried overnight at 60 °C, followed by the direct feeding in a twin-screw extruder (Leistritz ZSE 18 HP-40D) with the Ultranox 626A stabilizer for melt compounding under the processing conditions (the speed of screws was 200 rpm and temperature was in the range of 130–170 °C) [14,15]. The extruder was connected to a flat-heated die-and-calendering system to produce films of about 150 µm thickness.

For the preparation of three-layered films, triethoxycaprylylsilane-treated ZnO nanofiller, supplied by Umicore Zinc Chemicals (Angleur Belgium) as Zano 20 Plus, was used, with about 96% ZnO content. Joncryl ADR-4300F, supplied by BASF, a PLA chain extender with epoxy functionality (Mw = 5500, epoxy equivalent by weight 445 g/mol), was used at a percentage of 1% to limit the decrease in PLA molecular weights upon processing in the presence of ZnO particles [15].

In brief, the three-layered films were prepared following a 3-step method. First, a masterbatch of PLA and ZnO nanofiller with a content about 20 wt.% ZnO was prepared by melt-compounding (200 rpm and 130–170 °C), using a twin-screw extruder (Leistritz ZSE 18 HP-40D). After, the PLA and ZnO nanofillers were dried at 60 °C overnight and all the components were dry-blended in a mixer. The material coming out from the dye in the form of a rod was cooled down in a cold water bath, then granulated into pellets. Second, the masterbatch prepared in step one was diluted within a PLA matrix in the presence of ultranox 626A, by extrusion under similar conditions as melt-processing, to prepare pellets of the PLA–ZnO compound with a ZnO content of 1%. The masterbatch approach was used to promote and achieve good dispersion of ZnO nanofiller within the PLA matrix. Third, three-layered films composed of a PLA layer sandwiched between 2 layers of PLA–ZnO layers were produced from PLA and PLA–ZnO pellets by co-extrusion processing (Collin equipment). Each layer was around 50 µm thick and the overall film thickness was about 150 µm.


**Gentamycin-sulfate-loaded bioactive hydroxyapatite chitosan composite**


An electrophoretic deposition process was performed on pure titanium plates. Hydroxyapatite powder (particles <200 nm), chitosan (medium molecular weight 190–310 kDa, 75–85% deacetylation degree) and gentamycin sulfate solution (50 mg mL^–1^ in H_2_O) were used for obtaining composite coatings on Ti substrates (10 mm × 10 mm × 0.25 mm, 99.7% purity). All listed chemicals were purchased from Sigma-Aldrich. Before EPD, titanium plates were mechanically polished (grit emery paper and wet alumina) and degreased (15 min in acetone). Ti plates were kept in ethanol before proceeding with depositions.


**Electrophoretic deposition**


Composite HAP_CHI coating was deposited from an aqueous suspension containing 1 wt.% HAP powder and 0.05 wt.% chitosan in 1% acetic acid solution. HAP_CHI-GEN was deposited from an aqueous suspension containing 1 wt.% HAP powder, 0.05 wt.% chitosan and 0.1 wt.% gentamycin sulfate. The measured pH value was 4.4 for both suspensions. The suspensions were prepared as previously reported [16]. In brief, chitosan was dissolved in 1% acetic acid, and subsequently, the HAP powder was added under vigorous stirring and sonicated for 30 min. Thus, the resulting suspension contained 1 wt.% HAP and 0.05 wt.% CHI. HAP_CHI-GEN was prepared in the same manner, except for the addition of gentamycin sulfate solution, yielding a final suspension of 1 wt.% HAP powder, 0.05 wt.% chitosan and 0.1 wt.% gentamycin sulfate. EPD was performed as a cataphoretic deposition process on the Ti plate, serving as a working electrode (cathode) at previously determined conditions of 12 min deposition time and 5 V constant voltage [15]. As counter electrodes (anodes), two platinum panels were engaged. Uniform coating on both sides was achieved by placing the working electrode (Ti) equidistant at 1.5 cm between the Pt anodes. The deposited coatings were air-dried at room temperature for 24 h. The cross-sectional thicknesses of the as-deposited HAP_CHI and HAP_CHI-GEN coatings were 3.3 and 3.1 μm, respectively, measured by the polarized light microscope (Reichert MeF3) [17].

#### 2.2.3. Surface Sample Preparation

The control surface materials were first cleaned by immersing in 70% isopropanol containing 1 wt.% SDS and 1 wt.% lactic acid for 30 min, then rinsed with sterile distilled water and finally immersed in 70% isopropanol for a few seconds. Dried surfaces (glass token, 5 cent Euro coin or copper token) were sterilized by autoclaving. The cement or polymer surfaces containing various concentrations of zinc oxide, silver or antibiotics were disinfected under UV light for 1 h.

### 2.3. Inoculum Preparation

The *P. aeruginosa* ATCC 15442 and *S. aureus* ATCC 6538 colony-forming units (CFUs), arisen from a 24 h tryptic soy (TS) agar (TSA) culture, were transferred to a buffered sodium chloride–peptone solution, with a pH 7.0 (European Pharmacopoeia 10.5), to adjust inoculum suspension to 1 McFarland standard (approximately 3 × 10^8^ CFU/mL). This suspension was diluted 1/500 (test inoculum suspension) in a TS broth (TSB) medium enriched with 0.7 M NaCl (or in a brain–heart infusion (BHI) medium enriched with 2 wt.% glucose) for *S. aureus* and *P. aeruginosa*.

### 2.4. Biofilm Development Inhibition Assay after Transient Microbial Contact

A sterile 5 × 5 cm sterile gauze pad was placed in a petri dish of 10 cm diameter and was soaked with 20 mL of the test inoculum suspension. The cleaned and sterilized surface samples were placed on this soaked pad and after 5 min of direct contact, the samples were collected using sterile forceps and placed in a 12-well plate with the side in contact with the inoculum up (so not in contact with the 12-well plate). After 10 min of air drying under aseptic conditions, 1 mL of TSB enriched with NaCl or the brain–heart infusion broth (BHI) enriched with a glucose medium was added, the plate was covered and incubated at 37 °C to allow for full biofilm development. After 24 h, the microbial suspension and biofilm were collected from each well and the survival of microorganisms (CFU) was quantified. To achieve this, serial dilutions (1 in 10) of the well-retrieved microbial suspensions were prepared with the first dilution performed in an inactivating solution (2.57% of Letheen medium, 3.5% polysorbate 80, 0.33% lecithin, 0.1% L-histidine and 0.4% sodium lauryl sulfate) to eventually neutralize surface-released antimicrobial activities within 5 min. After a 5 min neutralization time, the dilution series were further pursued in peptone water. Of the three last dilutions, 800 µL was inoculated and well homogenized with the melted TSA medium (at 45 °C). The next steps included incubation of cooled-down dishes at 37 °C and CFU enumeration in appropriate plates after 24–48 h. Appropriate plates for counting were those with a CFU between 20 and 300 (or less in case low dilution plates showed <20 CFU) to obtain statistically significant data. Minor counting errors will have indeed more significant effects in the low-count range (<20 CFU) than in the higher range, unless the count range is too high (>300 CFU/9 cm Petri dish), leading to underestimation by CFU clustering. When possible, only plates with a CFU number between 20 and 300 should be enumerated. For well-retrieved microbial suspensions giving less than 20 CFU/plate for all dilutions, the CFUs on the plate corresponding to the less diluted samples were considered as “readable CFUs”. The following equation was used to determine the number of CFU/mL = 1.2 × (number of readable CFU × dilution factor of the well-retrieved microbial suspensions to obtain the plate with the readable CFUs). The tested surface antimicrobial activity was then assessed by calculating the logarithmic reduction (A) in the number of viable microorganisms (CFU/mL) on tested surface compared to the number of viable microorganisms (CFU/mL) on the negative control surface (without antimicrobial activity). The following formula was thus used: A = log (CFUc/CFUt), with CFUc corresponding to the number of CFU/mL obtained with the negative control surface, and CFUt corresponding to the number of CFU/mL obtained with the tested surface. All tests were performed at least three times in triplicates.

### 2.5. Biofilm Development Inhibition Assay with Persistent Microbial Inoculum Contact

The cleaned and sterilized surfaces were placed at the bottom of 12-well plate wells. Of the test inoculum suspension, 1 mL was added to each surface and the plate was incubated at 37 °C. After 24 h, both the planktonic microbial in the well and the biofilm were collected, and the survival of microorganisms (CFU) was quantified, as described in the upper Section 2.4. All tests were performed at least three times in triplicates.

### 2.6. Microbial Adhesion Assessment

After collecting the microbial and biofilm suspension in the 12-well plates during the biofilm development inhibition tests, surfaces were collected from the wells, immersed in 10 mL of buffered sodium chloride–peptone solution pH 7.0 for 2 min and placed in a new well, where bacteria were then detached from the surface by mechanical force (repeated pipetting). This was performed in an inactivating solution (2.57% of Letheen medium, 3.5% polysorbate 80, 0.33% lecithin, 0.1% L-histidine and 0.4% sodium lauryl sulfate) to neutralize any released antimicrobial compounds. The survival of microorganisms (which were previously adhering to the surfaces) was enumerated by CFU plate counting of serial sample dilutions, as described in the other biofilm development inhibition assays (see Section 2.4 and Section 2.5). All tests were performed at least three times in triplicates.

### 2.7. Biofilm Imaging

Glass control surfaces were treated using a protocol previously described [18]. Briefly, they were washed thrice in phosphate-buffered saline (PBS) and left in 2.5% glutaraldehyde and PBS for 24 h. Then, they were washed in PBS (×3), and left for 15 min in a series (50%, 70%, 90%, 96% and 100%) of ethanol solutions. Finally, the samples were immersed (×1) in hexamethyldisilazane and then left to dry for at least 10 min [18]. For SEM imaging, the samples were coated with platinum by standard procedures. Observations were made in a FEG-ESEM QUANTA F200 (FEI, TFS), using secondary electron imaging at 20–30 kV and a working distance of 8–11 mm. Images were analyzed by the AnalySIS iTEM 23082007 kjsoftware (Olympus, Münster, Germany).

### 2.8. Statistical Analysis

Statistical analyses were performed with the SPSS software (version 22.0, IBM Corp, New York, USA), using a one-way ANOVA test with a Tukey HSD post hoc test when analyzing the variance between the means of three or more groups, and using the Student’s test or the Wilcoxon–Mann–Whitney test when analyzing the variance between the means of two groups. Statistical significance was set to a *p <* 0.05.

## 3. Results

### 3.1. Repeatability and Robustness of Positive and Negative Control Surfaces in the Biofilm Development Inhibition Assay (Anti-Nascent Biofilm Assay) Performed with Persistent Microbial Inoculum Contact

In the first step, the selected control surfaces were validated regarding nascent biofilm inhibition, since all subsequent results are referenced to them. Therefore, we assessed the antibacterial activity of a 5 Euro cent coin (C) (antimicrobial activity positive control, copper-covered steel) and a borosilicate glass token (GT) (antimicrobial activity negative control) toward *Staphylococcus aureus* in the biofilm development inhibition test.

The experiments, which were performed six times in triplicates, showed that with an initial microbial inoculum mean of 2.3 × 10^5^ CFU/mL, the amount of the surviving *S. aureus* after 24 h incubation was about 1.8 × 10^10^ ± 1.8 × 10^10^ CFU/mL, while it was less than 1 × 10^3^ CFU/mL after contact with the 5 cent Euro coin. Therefore, the 5 cent Euro coin was an efficient antimicrobial, bactericidal, positive surface control. Indeed, a 7.02 ± 0.44 logarithmic survival reduction was obtained. In agreement with this result, a biofilm was observed on the glass token, while none was observed on the 5 cent Euro coin (Figure 1A) after incubation. Microscopic analyses of these surfaces after incubation confirmed the intact adhering bacteria developing a biofilm on the glass test coupon (Figure 1B), while the 5 cent Euro coin test coupons exhibited no attachment of bacteria. The CFU count confirmed the absence of anti-biofilm activity of the glass test coupon after 24 h with persistent contact with the microbial inoculum of 5.8 × 10^8^ CFU/mL. On the contrary, the number of surviving bacteria adherent on the 5 cent Euro coin was below the detection limit (<10 CFU/mL).

In the second step, the antibacterial activity of the control surfaces for this test was evaluated in parallel by two operators. The achieved logarithmic reductions of the surviving bacteria in the presence of a 5 cent Euro coin compared to the glass token were 6.7 and 7.3 for the two operators, confirming that the results were quite reproducible with no statistical difference in a student test (*p*-value = 0.1325).

In the third step, two procedures to disperse the collected bacteria after the 24 h incubation were investigated: bacterial dispersion was either performed in the well, by successive pipetting, or the whole culture medium was transferred to an Eppendorf tube and then dispersed by repeated pipetting. The results indicated that the differences were not significant (*t*-test, *p*-value = 0.69), suggesting that both the procedures are robust, since they lead to equivalent results.

The 5 cent Euro coin also consisted in an efficient antimicrobial positive surface control with regard to the nascent biofilm development not only for *S. aureus*, but also for the *P. aeruginosa* strain. It is worth noting that although the BHI medium enriched with glucose has been described as an efficient medium to induce *P. aeruginosa* biofilm growth development, the antimicrobial activity of the 5 cent Euro coin toward *P. aeruginosa* was reduced in this medium (2.6 ± 0.99 log reduction), compared to the test performed in TSB enriched with NaCl (12.59 ± 0.01 log reduction) (Figure 1C). The difference between the two logarithmic reductions was statistically significant (*p* < 0.05).

In agreement with these results, the copper token and zinc token tested under identical conditions, similarly reduced (>6 log reduction) the survival of *S. aureus* or *P. aeruginosa*.

### 3.2. Time-Dependent Antimicrobial Activity of the Biofilm Development Inhibition Test

We investigated the impact of different incubation times (5 min, 1 h, 4 h and 24 h) on the antimicrobial results obtained with 5 cent Euro coin assessed by persistent contact with the *S. aureus* inoculum. As shown in Figure 2, the antibacterial activity of the 5 cent Euro coin started after 1 h incubation (*p* < 0.05) and reached its maximum after 4 h. In contrast, bacteria continued their proliferation after 4 h of incubation in the wells containing glass token. This induced an increase in the logarithmic reduction over time (0.16 ± 0.15, 2.14 ± 0.12, 4.3 ± 0.32, 7.51 ± 0.21 log reduction after 5 min,1 h, 4 h and 24 h, respectively), even if a further reduction in the survival of bacteria could not be detected after 4 h in the presence of the 5 cent Euro coin.

### 3.3. Biofilm Development Inhibition Assay after Transient Microbial Inoculum Contact

Implant infections are supposed to be generally due to contamination, after transient contact with microorganisms during surgical procedures. Therefore, we assessed the efficiency of a biofilm development inhibition test, in which the surface is in contact with the bacterial suspension for only 5 min compared to the 24 h contact time for the inhibition test described above. CFU counting carried out after the 5 min transient contact with inoculum allowed to observe a reduction of approximately 14% CFU in the starting inoculum in the presence of the 5 cent Euro coin, compared to the condition in the presence of the glass token. This rapid bactericidal activity of the 5 cent Euro coin, allowing for a reduced starting inoculum, further increased the CFU logarithmic reduction in the assay performed after 5 min inoculum contact in the presence of this material, compared to the assay performed with 24 h contact (CFU log reduction = 8.8 ± 0.50).

### 3.4. Dose-Dependent Response and Flexibility in the Biofilm Development Inhibition Assay

The antibacterial activity of porous cement surfaces enriched with different percentages of silver were compared to that of the negative control in a biofilm development inhibition assay, with 24 h contact with *S. aureus*.

Although 0.0375 wt.% Ag cements were already significantly affecting bacteria survival (Figure 3), we were unable to demonstrate a dose response to the percentage of silver in the cements. Indeed, cements containing 1 wt.% Ag were not statistically more effective (Figure 3).

We also tested PLA (PolyLactic Acid) polymer surfaces with incorporated zinc oxide (ZnO), ZnOAg or silver through plasma technology, using our anti-nascent biofilm assay with persistent contact with *S. aureus*.

Amongst the nine surfaces tested, only ZnOAg-containing thin layers, having 80 nm thickness (PLA/80ZnOAg and PLA/3xZnO/80ZnOAg), showed a strong antibacterial activity (7 log reduction). These results suggested that while reproducible, the biofilm development inhibition test, robust and specific, could lack sensitivity, giving only significant bacteria log reduction in contact with a high dosage of antibacterial compounds (Figure 4).

Finally, we also investigated the anti-nascent biofilm activity of PCL-0.5% TET or PCL-1% TET compared to unloaded PCL surfaces on *S. aureus,* and observed more than a 6 log reduction after 24 h contact between the inoculum and the PCL-TET surfaces for both surfaces (Table 1). Gentamycin-loaded bioactive hydroxyapatite chitosan composite (HAP_CHI-GEN) also showed high (>4 log reduction) anti-nascent biofilm activities compared to unloaded HAP_CHI (Table 1).

## 4. Discussion

The described method is robust and simple, allowing for the differentiation of various material surfaces based on their anti-nascent biofilm properties. It only required basic bacteriology equipment (micropipettes, aseptic working conditions and a 37 °C incubator) and was taught easily, as shown during the training of summer school students, where training occurred in a few hours (iPROMEDAI COST). A test is carried out in 4 days. The first day, strain cultures are launched (5 min. work without medium preparation) and surfaces are cleaned and eventually sterilized or disinfected (surface preparation can be performed on day 2, if preferred). On day 2, surfaces are put in contact with the microbial inoculum, according to the chosen test (with persistent or transient contact). On day 3, surviving microorganisms are diluted and plated in adequate nutrient medium. Finally, on day 4, CFUs are counted.

The negative control surfaces, namely glass token, are adequate, since such material has no anti-nascent biofilm activity, as opposed to the metal elements from the positive control (5 cent Euro coin), reducing living bacteria by more than 6 log. The 5 cent Euro coin consists of copper-plated steel. The released copper can attack bacteria by damaging cell membranes, generating ROSs or degrading genomic and plasmid DNA [19]. However, when we compared its activity on different strains, our results suggested that the Gram-negative bacterium, *P. aeruginosa* ATCC 15442, was less susceptible. It is known that Gram-negative bacteria have more advanced detoxification systems than Gram-positive bacteria [20]. In a BHI medium with a high glucose concentration, the survival of the two bacterial strains was improved on contact with copper-releasing surfaces (results not shown for *S. aureus*). Indeed, Molteni et al. reported that the dissolution of copper ions can be affected by the contact medium composition [21]. Copper could also bind to certain components of the BHI medium, decreasing its availability. The age or previous usage of the coin had no impact on its antimicrobial activity, as we observed no variability in antimicrobial properties when using either new, used, or second-hand coins. However, the reuse of the 5 cent Euro coin in further biofilm development inhibition test could lead to decreased antimicrobial activities, as observed when reusing the 5 cent Euro coin positive control more than twice. Exceeding two subsequent disinfection procedures in 70% isopropanol, 1% SDS and 1% lactic acid for 30 min, followed by autoclaving, could affect the surface of the coin, reducing copper release either by affecting surface oxidation, corrosion or stabilization. The dodecyl sulfate could also bind to Cu and Zn instead of the sodium-reducing free Cu and Zn at the surface. Nevertheless, Cu and Zn are oxidized during autoclaving, and pure Cu is probably consequently replaced by copper(II) oxide (CuO), copper(I) oxide (Cu_2_O) and Cu(0) metallics that are known for their antimicrobial properties [22].

The robustness of the techniques and controls (positive and negative) were also satisfactory as the introduction of variations (different manipulators with low or no skills in bacteriology, different sampling procedures, different bacterial strains, different contact time) did not significantly modify the results.

The assays were specific, since each test (with persistent microorganism contact or transient microorganism contact or the adhesion test) led to different activities, allowing for eventually improving the surface activity understanding. Indeed, the assay performed with a persistent inoculum highlights the antibacterial and anti-nascent biofilm activity of the surface, mainly for surfaces releasing antimicrobial activities. The assay performed with short inoculum contact better demonstrates an anti-adhesion and antimicrobial activity of contact (even within 5 min), since only surviving bacteria adhering to the surface will replicate in the well. Note that bacteria may be detached from surfaces at any time during the assay, allowing for counting adherent living bacteria. This bacteria adhesion test allows not only to assess microbe adhesion efficiency, but also to assess the anti-nascent biofilm activity after 6 h incubation. This anti-nascent biofilm activity can be the consequence of different antibacterial activity (biocidal, biostatic and/or an anti-adhesion surface activity). The proposed assays meet our initial objective, which was to highlight the anti-microbial activity of each implant surface, regardless of its mechanism of action. Nevertheless, the anti-nascent biofilm assessment can only be carried out after 6 h of incubation, the time generally required for biofilm formation [23]. By scanning electron microscopy, although surfaces were dehydrated for microscopy procedures, we could observe biofilm structure on the negative control surface after 24 h of bacterial culture. On the contrary, on the 5 cent Euro coin surface, bacteria were absent or dead (absence of biofilm). This result was expected as copper can kill bacteria within 12 min to 6 h, depending on the contact medium [21]. This further highlights the potential of Cu-coated surfaces as antimicrobial alternatives, especially in the hospital environment. Sample surfaces, having a reduced bactericidal potential compared to copper, lead to reduced anti-nascent biofilm activities.

The lowest dose of detected antibacterial activity (sensitivity) depends on the composition of the surface. A cement with 0.0375 wt.% Ag could reduce the number of living bacteria by 7.37 log, while 5 nM ZnOAg in polylactic acid barely achieved one log reduction (one order of magnitude). However, the results obtained with surfaces containing various concentrations of those antimicrobial compounds suggested a short measuring interval with detection linearity. Indeed, we did not observe a gradual decrease in living bacteria with increasing doses of antimicrobial compounds on the surface. Our incubation time may have been too long, resulting in too many bacteria generations to obtain the same efficiency of the antibacterial effect during the incubation time. Therefore, when the assay aims at observing a dose-dependent antibacterial activity, a shorter time (e.g., 6 h) should be applied to keep a relatively low target bacteria amount to assess surface antibacterial activities. It is worth noting that in another study, lower antimicrobial activity (on *S. aureus* and *S. epidermidis*) could only be detected using 0.01 wt.% Ag to 0.00375 wt.% Ag [24].

The 20 cent Swiss franc and the 10 pence British coin, both composed of 25% copper and 75% nickel, displayed similar anti-nascent biofilm activities as the 5 cent Euro coin. Copper and zinc tokens also showed high anti-nascent biofilm activities (>4 log reduction). Coins can only be intended as positive antimicrobial surfaces, without consideration of complex architectures. Specific additional control with complex architecture can be added when necessary.

We were able to test different surfaces, porous or non-porous, composed of different antimicrobial agents, including cement surfaces releasing Ag^+^, as well as PLA polymers associated with ZnO and/or Ag^+^. Porous cement surfaces releasing Ag had a strong antibiofilm and antibacterial activity as the ZnOAg–PLA, on the opposite-to-ZnO–PLA or Ag–PLA surfaces. Ag^+^ and Cu^2+^ ions can induce irreversible cell damage [25]. Surfaces releasing antibiotics, such as HAP_CHI-GEN or PCL-TET, also showed clear (>4 log) anti-nascent biofilm activities. Our assay can thus be applied to assess porous and non-porous surfaces. As proof of the concept, a new analysis of polyurethanes with antibacterial properties confirms the reliability of this approach [26,27].

It is worth noting that the antifungal or antibiofilm activities from fungi proliferation could also be used in these assays/methods. Indeed, for assessing the effectiveness of antimicrobial surfaces, it would be useful to test their activities against three biofilm-developing strains, having a different cell wall (external membrane or not) or metabolism and cell replication (prokaryote or eukaryote): a Gram-positive strain (*S. aureus*, such as ATCC 6538), a Gram-negative strain (*P. aeruginosa*, such as ATCC 15442) and a yeast (*C. albicans*, such as ATCC 10231, not included in this study). In addition, although we performed our assays in static conditions, our procedure allows us to eventually add a shear stress using orbital agitation during incubation. This could allow for dynamic biofilm studies under controlled fluid conditions [28].

Finally, it is also possible to assess antimicrobial activities of surfaces on nascent multispecies biofilms. The assay could, in that case, be performed using an inoculum made up of equal mixtures of *S. aureus, P. aeruginosa* and *C. albicans*, with CFU enumeration performed in parallel on a cetrimide medium (selective toward *P. aeruginosa*), on a Chapman medium (selective towards *S. aureus*) or a Sabouraud medium with gentamycin (selective toward *C. albicans*).

## 5. Conclusions

In summary, we present here robust, specific and sensitive methods to easily identify anti-nascent biofilm surfaces. The reliable and commercially available negative and positive controls could boost international comparisons not only across Europe, but also globally. These flexible methods can assess antimicrobial activity of porous and non-porous surfaces, regardless of their mechanism of action.

## Figures and Tables

**Figure 1 microorganisms-11-01023-f001:**
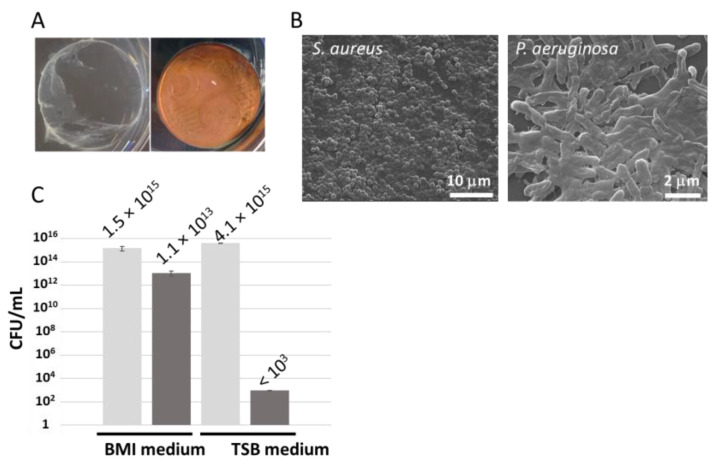
(**A**) Visual aspect of surfaces (glass token: left; 5 cent Euro coin: right) after 24 h incubation with *S. aureus*. (**B**) Microscopy analysis of glass tokens after 23 h incubation with *S. aureus* (left) and *P. aeruginosa* (right). (**C**) Impact of the growth medium (BHI or TSB) on the biofilm development inhibition test performed in persistent contact with the *P. aeruginosa* inoculum. The CFU/mL obtained with negative control glass surface or positive control 5 cent Euro coin are shown in light grey and dark gray, respectively.

**Figure 2 microorganisms-11-01023-f002:**
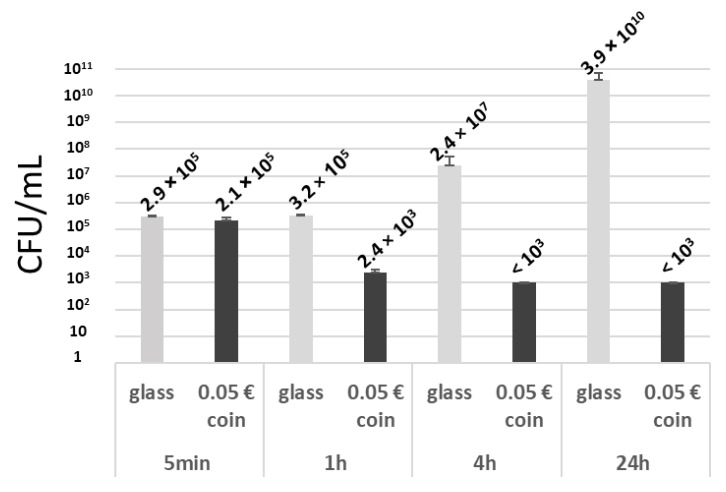
Level of surviving bacteria (CFU/mL) after contact during various time periods (5 min, 1 h, 4 h and 24 h) with the negative control glass surface (light grey) or the 5 cent Euro coin (dark grey) positive control.

**Figure 3 microorganisms-11-01023-f003:**
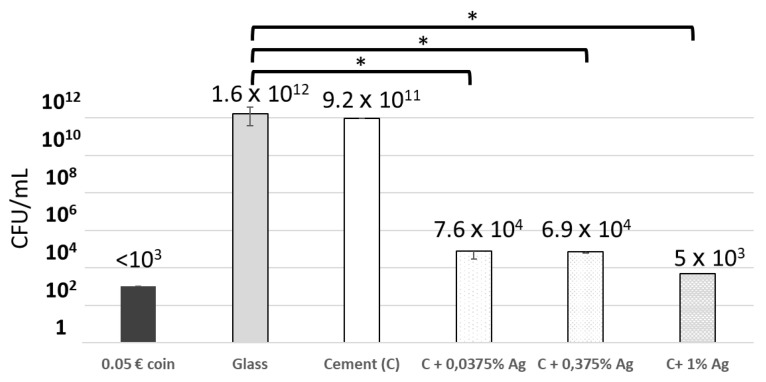
Surviving *S. aureus* (in CFU/mL) after 24 h contact with cements enriched with different silver percentages. Test carried out five times (* *p*-value < 0.05).

**Figure 4 microorganisms-11-01023-f004:**
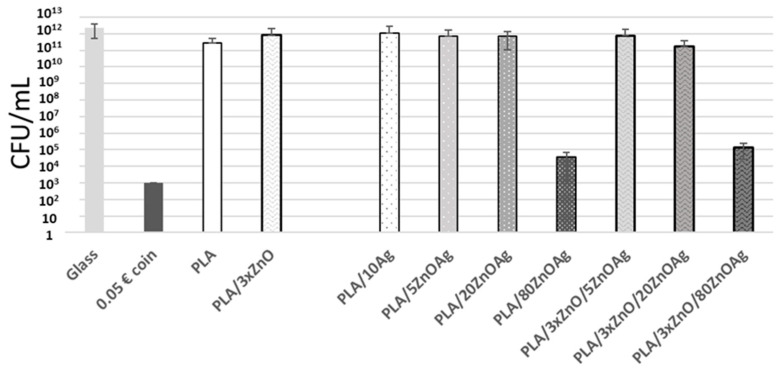
Surviving *S. aureus* (in CFU/mL) after 24 h contact with different surfaces: glass (light grey column), 0.05 Euro coin (dark grey column), PLA (white column), eventually including a 10 nm plasma Ag layer (PLA/10Ag), (white column with low amount of dots) or PLA/ZnO monolayer including 5 nm (light grey column with low amount of dots), 20 nm (light grey column with medium amount of dots), 80 nm (dark grey column with high amount of dots) ZnOAg (PLA/5ZnOAg, PLA/20ZnOAg, PLA/80ZnOAg) and PLA/ZnO trilayer 1% (white column with “waves”), eventually including 5 nm (light grey column with waves), 20 nm (grey column with waves) or 80 nm (dark grey column with waves) plasma ZnOAg layer (PLA/3xZnO/5ZnOAg, PLA/3xZnO/20ZnOAg, PLA/3xZnO/80ZnOAg). Tests were carried out three times per surface.

**Table 1 microorganisms-11-01023-t001:** Antibacterial activity of the tested surfaces with *S. aureus*.

Material	Log Reduction of Bacterial Inoculum at 24 h
Glass test coupon	0
5 cent Euro coin	7
Cu test coupon	>6
Zn test coupon	>6
PLA/80 nm plasma ZnOAg	7
PLA/ZnO trilayer 1%/80 nm plasma ZnOAg	7
HAP_CHI-GEN	>4
PCL-0.5% TET	>6
PCL-1% TET	>6

## Data Availability

Data are available at https://doi.org/10.5281/zenodo.7753718, accessed on 10 April 2023.

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
