# Peer review of "Easy, Flexible and Standardizable Anti-Nascent Biofilm Activity Assay to Assess Implant Materials"

_microorganisms, 2023, doi:10.3390/microorganisms11041023_

Round 1

Reviewer 1 Report (Previous Reviewer 2)

Dear Authors: 

I reviewed your manuscript with great interest.  However, you didn't explain the background enough in your introduction part. 

1. Please refer to the following book.  Kanematsu, Hideyuki, and Dana M. Barry. "Formation and control of biofilm in various environments." (2020).  They introduced some standards already set in the past and outlined the new standardization of biofilms evaluation on materials.  Or their review paper appeared in International Materials Review:  Kanematsu, Hideyuki, Dana M. Barry, Hajime Ikegai, and Yoshimitsu Mizunoe. "Biofilm control on metallic materials in medical fields from the viewpoint of materials science–from the fundamental aspects to evaluation." International Materials Reviews (2022): 1-25.   Please check those reviews and clarify the standpoint of your evaluation method in the world's new trend.  Please rewrite and revise your introduction part and also the discussion part.  

2. Your evaluation method seems to be based on the CFU measurement.  The method makes the difference between antibacterial evaluation and biofilm vague.  You should explain the difference in the discussion part clearly.  Probably, you think that the number of bacteria must play an essential role in the initial stage of biofilm formation.  If so, I agree with you, but the CFU should not be measured in the bacterial solution but on the material's surface.  Usually, the measurement would be complicated.  Could you consider it?  If not, please revise your discussion part and conclusion properly.   

Author Response

Reviewer 2 Report (Previous Reviewer 3)

The authors have modified the manuscript based on the comments given by the reviewers. However, I have the following minor comments to address before the manuscript can be considered for publication.

1. Please check the sub-title numbering of 'Materials and Methods' section.

2. Please make the strain names in italics throughout the manuscript.

Author Response

Reviewer 3 Report (Previous Reviewer 4)

A simple and relatively straight forward paper  that could benefit from the following improvements:

1.     Author should provide a graphical abstract to increase appeal.

2.       English language must be revised. I suggest that the manuscript by reviewed by a native speaker for language mishaps.  

Round 2

Reviewer 1 Report (Previous Reviewer 2)

I can confirm the authors sincerely revised their original manuscript.  Hopefully, this revised manuscript will appear in the journal.  

This manuscript is a resubmission of an earlier submission. The following is a list of the peer review reports and author responses from that submission.

Round 1

Reviewer 1 Report

The publication is well written, unfortunatly the take hoem message is very low.

The publication title does not reflect the subject of the results.

The microbiologic tests used are basic ones. There is no new point of view.

All the researchers that are dealing with antibacterial activity are using the same approach. Spending more the 50% of the paper saying that 5 cents euro coins can be a positive control of an antibacterial activity is superfluous.  It is already known. Furthermore, when it is necessary to compared with complex architectures such as those presented or suggested in the paper, theses coins are not the best antimicrobial positive control.

The results, as presented here, do not have any added value.  Some of them could be highlighted but with experiments making it possible to understand the mechanisms underlying the antibacterial and anti-biofilm activity of these structures.

In the discussion the first part is more a material and methods section, in any case it is not scientifically relevant. After this part, the copper activity is discussed but only with the help of the literature. No assay or results in the paper are supporting the message.

What is the rational for using and comparing Silver-loaded cements , Monolayered (PLA/Ag or PLA/ZnOAg) and three-layered ZnO PLA surfaces , HAP_CHI, HAP_CHI-GEN with 5 cents euro coins antibacterial activity. All the results shown are “preliminary”  or “end point” data and should be combined with physical, chemical and mechanistic approaches, to have a take home message.

Author Response

We thank the reviewers for their comments. We introduced changes accordingly and we believe that those improved the quality of our manuscript.

Please find here our response to reviewer 1's comments:

  • The publication is well written, unfortunatly the take hoem message is very low.”

The texts from the Abstract, Introduction and Discussion have been improved to have a clearer take home message. See the revised manuscript.

  • The publication title does not reflect the subject of the results”.

Title has been improved.

  • The microbiologic tests used are basic ones. There is no new point of view”.

Indeed, tests are easy but robust to allow international comparison. The main take home message is to describe a simple and robust method and also accessible control items to perform implant surface antimicrobial activity assessment. Hopefully, this take home message is now clearer in the revised manuscript.

  • All the researchers that are dealing with antibacterial activity are using the same approach. Spending more the 50% of the paper saying that 5 cents euro coins can be a positive control of an antibacterial activity is superfluous. It is already known”. “Furthermore, when it is necessary to compared with complex architectures such as those presented or suggested in the paper, theses coins are not the best antimicrobial positive control”.

One of the main goal of the iPROMEDAI Cost action (more than 100 members) was to deliver such an assay and controls with rigorous description of the assay conditions to allow standardization and international comparison, obviously because industries and academic researchers were lacking this type of assays. Coins are only intended as positive antimicrobial surfaces in a standardizable assay without consideration of complex architectures (as it should be for general positive control).  No previous reports used coins as positive controls. Specific control with complex architecture can be added. This is now added lines 740-742 (509-512) in the discussion.

- “The results, as presented here, do not have any added value.  Some of them could be highlighted but with experiments making it possible to understand the mechanisms underlying the antibacterial and anti-biofilm activity of these structures.”.

The results present the characteristics of the assay: easiness, robustness, flexibility, etc. The mechanisms explaining the antimicrobial activities of the various surfaces is not the goal of this manuscript. We hope to make the clearer the take home message.

In the discussion the first part is more a material and methods section, in any case it is not scientifically relevant.”

Description and validation of a relevant method to assess easily the anti-nascent biofilm activity of implant surfaces could allow international comparison of the various published results.  It will be useful for industries and researchers dealing with surfaces intended for implants. It is thus scientifically relevant.

“After this part, the copper activity is discussed but only with the help of the literature. “

As the activity of the copper is already described and understand, we believe that that scientific literature is sufficient.

No assay or results in the paper are supporting the message”.

Our “Material and Methods” and “Results” describe the assays and the various results and characteristics of the assays obtained with different surfaces, including coins and token with copper.

"What is the rational for using and comparing Silver-loaded cements , Monolayered (PLA/Ag or PLA/ZnOAg) and three-layered ZnO PLA surfaces , HAP_CHI, HAP_CHI-GEN with 5 cents euro coins antibacterial activity. All the results shown are “preliminary”  or “end point” data and should be combined with physical, chemical and mechanistic approaches, to have a take home message".

We improve the take home message so that it is now clear (see the revised manuscript). The rational for using this various surfaces was intended to show the flexibility of the assay, that could not only assess non-porous materials but also porous materials. This is now better described in the revised manuscript. Our goal is not to assess implant surfaces by physical, chemical and mechanistic approaches. However, we add a recent reference (ref. 19) on the silver-loaded cements.

Reviewer 2 Report

Dear authors:

I read your well written article with great interest.  This article might be well accepted as antibacterial paper.  However, I would say that your experiments could not show antibiofilm evaluation.  I mean, you mixed antibacterial evaluation and antibiofilm one without any distinct demarcation.  Therefore, I would like to ask you the following three major points.

1. Please clarify the definition of each technical term, so that the differences would be shown  more clearly.  The explanation could be placed in introduction part, for example.  

2. Please show the existence of biofilms by other methods, for example, optical microscope, SEM-EDX, Confocal laser microscope, AFM, Raman spectroscopy, FT-IR.  

3. Biofilms could not be evaluated only by colony formatoin units nor by bacterial numbers.  Therefore, you should select other methods.  

4. If you would feel the revision impossible, please delete antiboifim evaluation and concentrate on antibacterial characteristics.  In such a case, there is already a very powerful international standard for antibacterial measurement.  Please compare your method to the ISO.  See: 1. Kanematsu, Hideyuki. "A New International Standard for Testing Antibacterial Effects." Advanced Materials & Processing 175, no. 4 (2017): 26-29.

5. On the other hand, SIAA (Japanese industrial organization for antibacterial articles) is going to establish ISO very soon.  For your reference:

Kanematsu, Hideyuki, Dana M Barry, Hajime Ikegai, and Yoshimitsu Mizunoe. "Biofilm Control on Metallic Materials in Medical Fields from the Viewpoint of Materials Science–from the Fundamental Aspects to Evaluation." International Materials Reviews (2022): 1-25.

Please note that the biofilm formation process is composed of muti steps.  The attachment of bacterial is related just to one of the steps and would affect the quroum sensing, that's for sure, but this doesn't be always lead to antibiofilm evaluation.  

And this is a minor revision which I would like to claim.  You should include the experimental results and conclusion in the abstract section simply.  

Author Response

We thank the reviewers for their comments. We introduced changes accordingly and we believe that those improved the quality of our manuscript.

Please find here changes introduced in regards to Reviewer 2 comments:

  1. "Please clarify the definition of each technical term, so that the differences would be shown  more clearly.  The explanation could be placed in introduction part, for example."

Thank you for your comment. We clarified our goals, some definitions. This is now in the title and introduction. 

2. "Please show the existence of biofilms by other methods, for example, optical microscope, SEM-EDX, Confocal laser microscope, AFM, Raman spectroscopy, FT-IR. "

The existence of biofilms is shown visually (Figure 1A) and with the electronical microscope for both S. aureus and P. aeruginosa biofilms (Figure 1B).

3. "Biofilms could not be evaluated only by colony formation units nor by bacterial numbers.  Therefore, you should select other methods."

We do not intend to assess the antibiofilm activity on mature biofilm as those are not present on medical devices or implants since these surfaces need to be sterile at the beginning. However, during the surgical or implantation processes, microorganisms can be in contact with the surfaces, bind surfaces and develop biofilms which are composed of microorganisms and their secreted compounds. When bacteria are killed, biofilm is also not anymore produced. So if bacteria are directly killed they won’t be bacteria and biofilm at the surface. We observed this fact in our results with electronic microscopy. Therefore, the main goal of the antimicrobial activities of surfaces intended for implants is to avoid infections which start during surgery or implantation. Surface activities impeding bacteria growth, survival and adherence are therefore targeted. Implant surface antibiofilm activities on mature biofilms are not targeted. This is why here we are showing that the method and controls allow for analyzing those specific antimicrobial activities on nascent biofilm (“biofilm development inhibition test):  reducing microbial development, attachment and consequently biofilm development. This is now clearly mentioned in the introduction and title.  

Kanematsu et al. referenced by the reviewer also clearly mentioned in his review “Antibacterial effects of metallic materials must lead to a decrease in the number of bacteria on materials’surfaces. Therefore, this effect contributes to antibiofilm properties ((b) in Figure 3). In light of that, antibacterial effect of metallic ions would lead to antibiofilm properties of materials”. However they further mentioned “However, biofilm are not evaluated only by antibacterial effects alone, since the killed and collapsed bacteria could be one of the components in biofilms. This is the reason why the number of living bacteria could not precisely evaluate the antibiofilm properties of materials”.  Indeed, this is true when assessing activities against mature biofilm. However, this is not the case for medical devices or implants as the surfaces are sterile before use and biofilm only arises from a living microorganism. So, biofilm on a medical device or implant always starts from one living microorganism that will subsequently develop biofilm through a multistep process.

4. "If you would feel the revision impossible, please delete antiboifim evaluation and concentrate on antibacterial characteristics.  In such a case, there is already a very powerful international standard for antibacterial measurement.  Please compare your method to the ISO.  See: 1. Kanematsu, Hideyuki. "A New International Standard for Testing Antibacterial Effects." Advanced Materials & Processing 175, no. 4 (2017): 26-29."

We have improved the title but also the introduction, Results and Discussion sections and we hope we are clearer about our definitions and claims. Thank you for this ISO information. However, we could not find it. We are therefore referring to the ISO 22196, the review and the book of Kanematsu and Barry in the introduction.

  1. "On the other hand, SIAA (Japanese industrial organization for antibacterial articles) is going to establish ISO very soon.  For your reference: Kanematsu, Hideyuki, Dana M Barry, Hajime Ikegai, and Yoshimitsu Mizunoe. "Biofilm Control on Metallic Materials in Medical Fields from the Viewpoint of Materials Science–from the Fundamental Aspects to Evaluation." International Materials Reviews (2022): 1-25."

Thank you for this interesting reference. This reference is now introduced in our introduction.

6.  "Please note that the biofilm formation process is composed of muti steps.  The attachment of bacterial is related just to one of the steps and would affect the quorum sensing, that's for sure, but this doesn't be always lead to antibiofilm evaluation. "

We are working quite extensively on biofilm (see for example Cpn60.1 (GroEL1) Contributes to Mycobacterial Crabtree Effect: Implications for Biofilm Formation. Zeng S, Constant P, Yang D, Baulard A, Lefèvre P, Daffé M, Wattiez R, Fontaine V. Front Microbiol. 2019 Jun 11;10:1149. doi: 10.3389/fmicb.2019.01149. ) and this multistep process is described lines 62-77 in our manuscript.

Minor revision: "And this is a minor revision which I would like to claim.  You should include the experimental results and conclusion in the abstract section simply. "

We modified the sentences about our experimental results and conclusion in the abstract, hopefully to improve the take-home message.

Thank you for your help.

Reviewer 3 Report

In this article, the authors have come up with a robust and simple assay method to identify antimicrobial and antibiofilm surfaces. However, the authors should re-organize the manuscript giving emphasis to the language to make the readers understand the content easily. It would be better to write the content more concisely and clearly. In addition, the reference needs to be included for every fact explained in the manuscript.

Some of the comments to be addressed are given below.

L35-38: Please re-write the sentence

L66: depends

L73-77: Please re-write it

L129-130: Change bold to normal fonts. Check it throughout the manuscript

L170, 171, 177, 185: Change bold to normal fonts

L188: by immersing

L190: immersed in 70% isopropanol for few seconds.

L192: various concentration of

L201: Remove ‘suspension’

L205: after 5 min of direct contact

L207: After 10 min of air drying

L210: Re-write the sentence

L219: 24 – 48 h.

L196: 24 h

L219: Appropriate plates

L220: CFU between

L224-226: Re-write the sentence

L242: chapter à section

L240: both the

L241: survival of microorganisms

L242: See 3.3 section à No need to indicate results sessions in the Materials and Methods

L247: Replaced in a new well à Re-write it

L251: Survival of microorganisms

L253: see 3.3 and 3.4 sections à No need to indicate results sessions in the Materials and Methods

L305: Re-write the sentence

L308: repeated

L308: Mention it as data not shown

L315: P. aeruginosa à italics

L320: Please re-write the sentence

L321: italics for bacterial names. Correct it throughout the manuscript

L329: 1 h, 4 h and 24 h: Correct the spacing throughout the manuscript

L330: survival of bacteria

L348-351: Need to re-write

Result section 3.3: No data mentioned in the text

L395-398: Need to re-write it

Author Response

We thank the reviewers for their comments. We introduced changes accordingly and we believe that those improved the quality of our manuscript.

Please find here below the changes introduced in regards to reviewer 3 comments : "In this article, the authors have come up with a robust and simple assay method to identify antimicrobial and antibiofilm surfaces. However, the authors should re-organize the manuscript giving emphasis to the language to make the readers understand the content easily. It would be better to write the content more concisely and clearly. In addition, the reference needs to be included for every fact explained in the manuscript".

 We deleted some sentences to be more concise and add some information to be clearer (see the Revised Manuscript). Six references have been added:

  • Kanematsu, H.; Barry, D.M.; Ikegai, H.; Mizunoe, Y. Biofilm control on metallic materials in medical fields from the viewpoint of materials science – from the fundamental aspects to evaluation. Mater. Rev. 2022, 1-25.
  • Kanematsu, H.; Barry, D.M. Formation and Control of Biofilm in Various Environments, 1st. ed.; Springer: Singapore, 2020. https://doi.org/10.1007/978-981-15-2240-6
  • Castro, J.; Lima, Â.; Sousa, L. G. V.; Rosca, A. S.; Muzny, C. A.; Cerca, N. Crystal Violet Staining Alone Is Not Adequate to Assess Synergism or Antagonism in Multi-Species Biofilms of Bacteria Associated With Bacterial Vaginosis. Cell. Infect. Microbiol. 2022, 11, 795797. https://www.frontiersin.org/articles/10.3389/fcimb.2021.795797. DOI=10.3389/fcimb.2021.795797
  • Azeredo, J.; Azevedo, N.F.; Briandet, R.; Cerca, N.; Coenye, T.; Costa, A.R.; Desvaux, M.; Di Bonaventura, G.; Hébraud, M.; Jaglic, Z.; Kačániová, M.; Knøchel, S.; Lourenço, A.; Mergulhão, F.; Meyer, R.L.; Nychas, G.; Simões, M.; Tresse O.; Sternberg, C. Critical review on biofilm methods. Rev. Microbiol. 2017, 43:, 313-351, DOI:10.1080/1040841X.2016.1208146
  • Jacquart, S.; Girod-Fullana, S.; Brouillet, F.; Pigasse, C.; Siadous, R.; Fatnassi, M.; Grimoud, J.; Rey, C.; Roques, C.; Combes, C. Injectable bone cement containing carboxymethyl cellulose microparticles as a silver delivery system able to reduce implant-associated infection risk. Acta Biomater. 2022, 145, 342-357. doi: 10.1016/j.actbio.2022.04.015. 
  • Gomes, J.C.; Mergulhão, F.J.M. A Selection of Platforms to Evaluate Surface Adhesion and Biofilm Formation in Controlled Hydrodynamic Conditions. Microorganisms. 2021, 9, 1993. https://doi.org/10.3390/ microorganisms9091993

Thank you also for your comments. We improved our manuscript accordingly.

- "L35-38: Please re-write the sentence."  We modified the sentences about our experimental results and conclusion in the abstract.

- "L66: depends". Corrected.

- "L73-77: Please re-write it". Done.

- "L129-130: Change bold to normal fonts. Check it throughout the manuscript". Done

- "L170, 171, 177, 185: Change bold to normal fonts". Done

- "L188: by immersing". Done.

- "L190: immersed in 70% isopropanol for few seconds". Done.

- "L192: various concentration of." Done.

- "L201: Remove ‘suspension’. Done.

- "L205: after 5 min of direct contact". Done.

- "L207: After 10 min of air drying". Done.

- "L210: Re-write the sentence". Done.

- "L219: 24 – 48 h". Done.

- "L196: 24 h". Done.

- "L219: Appropriate plates". Done.

- "L220: CFU between". Done.

- "L224-226: Re-write the sentence". Done.

- "L242: chapter à section". Done.

- "L240: both the". Done.

- "L241: survival of microorganisms." Done

- "L242: See 3.3 section à No need to indicate results sessions in the Materials and Methods,". Numbers of sections corrected.

- "L247: Replaced in a new well à Re-write it". Done.

- "L251: Survival of microorganisms." Done.

- "L253: see 3.3 and 3.4 sections à No need to indicate results sessions in the Materials and Methods." Numbers of sections corrected.

- "L305: Re-write the sentence." Done.

- "L308: repeated." Done.

- "L308: Mention it as data not shown." Done.

- "L315: P. aeruginosa à italics" Done.

- "L320: Please re-write the sentence".  Done.

- "L321: italics for bacterial names. Correct it throughout the manuscript." Done.

- "L329: 1 h, 4 h and 24 h: Correct the spacing throughout the manuscript." Done.

- "L330: survival of bacteria". Done.

- "L348-351: Need to re-write". Done.

- "Result section 3.3: No data mentioned in the text. Data are mentioned but not shown". This is now mentioned.

- "L395-398: Need to re-write it". Done.

Reviewer 4 Report

The manuscript must be revised my a native English speaker. The Abstract and Introduction are riddled with grammar issues that make the text barely comprehensible. 

Unfortunately, the paper as a whole is written in a very cryptic expression making it very hard to read. 

I recommend the addition of a graphical abstract to increase appeal to readers. Also, consider presenting more tables and better charts and graphs. 

Efforts must be made both in the presentation of results and also the formulation of text. 

References are scarce, I recommend making more comparisons with results from previously published papers on the similar subjects.  This would immensely benefit the Discussion part.  

Last, but not least, there is something to be said about the substantial number of authors. The number of authors seems to be higher than the number of pages and roughly equal to the number of cited references. 

In conclusion, I do not support the publication of this paper in its present form. 

Round 2

Reviewer 2 Report

Dear authors:

I confirmed that you sincerely did your best to improve the original version and also felt that the contents were improved significantly.  I hope that your manuscript will be accepted and published in the near future.  

Reviewer 3 Report

The revised manuscript has been improved drastically. However, the authors should modify the manuscript giving emphasis to scientific writing, sentence structure, spacing, usage of comma etc. Some of the comments to be addressed are given below.

L98: Sterile before use? Please modify the sentence

L109: crystal violet

L151: Petri dishes à petri dishes

L189: speed of screws=200rpm à speed of screws = 200 rpm

L189: 130-170°C) à 130-170 °C) (Please check the spacing throughout the manuscript before °C)

L250: 24h à 24 h (Please check this throughout the manuscript)

L388: Please re-write the sentence

Figure 2 legend: remove ‘, respectively’.

Figure 3: CFU indication à please include ‘×’. E.g. 1.6 × 1012

L527: observed more than 6 log reductions after 24 h contact between the inoculum and the PCL-TET for both surfaces

L530: also showed

Table 1: 24 h

L541: were instructed within hours à Please re-write the sentence

L554: bacterium

L554: was less

L556-558: Please re-write the sentence

L610: within 12 min to 6 h

L610-612: Please modify the sentence

L622: shorter time (e.g. 6 h)

L633: Combine with the paragraph above

L635: During these manipulations? Please modify the sentence

L657-658: Please re-write the sentence

L656: also showed clear

L675: gentamycin
